**communications** engineering

# Fully automated multicolour structured illumination module for super-resolution microscopy with two excitation colours

Haoran Wang[1], Peter T. Brown [2], Jessica Ullom[2], Douglas P. Shepherd[2], Rainer Heintzmann [1,3] & Benedict Diederich [1] ✉

In biological imaging, there is a demand for cost-effective, high-resolution techniques to study dynamic intracellular processes. Structured illumination microscopy (SIM) is ideal for achieving high axial and lateral resolution in live samples due to its optical sectioning and low phototoxicity. However, conventional SIM systems remain expensive and complex. We introduce openSIMMO, an open-source, fully-automated SIM module compatible with commercial microscopes, supporting dual-color excitation. Our design uses affordable single-mode fiber-coupled lasers and a digital micromirror device (DMD), integrated with the open-source ImSwitch software for real-time super-resolution imaging. This setup offers up to 1.55-fold improvement in lateral resolution over wide-field microscopy. To optimize DMD diffraction, we developed a model for tilt and roll pixel configurations, enabling use with various low-cost projectors in SIM setups. Our goal is to democratize SIM-based super-resolution microscopy by providing open-source documentation and a flexible software framework adaptable to various hardware (e.g., cameras, stages) and reconstruction algorithms, enabling more widespread super-resolution upgrades across devices.

Fluorescence microscopy is a fundamental technique for exploring biological processes at the micrometer scale which has revolutionized our understanding of the life sciences. However, our understanding of many important biological processes is limited by tradeoffs in fluorescence microscopy which make it challenging to image large volumes at high-resolution and high speeds. Recently, a wide variety of advanced imaging instrumentation has been developed to mitigate these tradeoffs, but these approaches often require significant technical expertise to build and operate or are prohibitively expensive for most life science groups. We address this problem for fluorescence super-resolution imaging by designing a low-cost, open-source add-on for realizing structured illumination microscopy on commercial microscope stands.

The demands, especially on optical imaging, are increasing in order to depict finer structures, larger volumes and faster dynamics, for example to better understand the interactions between virus and host or to recognize the effects on our inner immune system[1]. This often leads to an increase in the required technical expertise to build or operate a recently developed device or, considering commercial devices, an extremely high purchase price, whereby existing equipment can often not be reused.

To increase the optical resolution beyond the diffraction limit defined by Abbe, numerous research groups have developed innovative approaches over the past two decades. These include localization microscopy techniques based on photo-switching such as $d$STORM[2,3] and PALM[4], as well as point-scanning approaches that rely on saturated molecular transitions such as STED and RESOLFT[5]. However, these approaches exploit fluorophore photophysics and thus require the selection of a specific fluorescent dye and high illumination intensity, which can lead to phototoxicity or photobleaching, all of which represent major hurdles for live cell imaging[6].

Structured illumination microscopy (SIM[7,8]), as a widefield imaging technique, typically offers a maximum improvement in resolution by a factor of 2. It requires lower excitation intensity, making it a gentler method for imaging living cells, while keeping the amount of data necessary for the resolution improvement small. This makes it ideally suited for observing dynamic cellular processes with minimal photodamage.

SIM achieves its high resolution by illuminating fluorescent samples with a series of very fine periodic patterns of very high contrast and detecting images at various pattern positions for computational reconstruction. This structured illumination down-modulates high spatial frequencies of the object into the detection passband of the microscope and the computational

[1]Leibniz Institute of Photonic Technology, Albert-Einstein-Str. 9, 07745 Jena, Germany. [2]Arizona State University, Center for Biological Physics and Department of Physics, Tempe, AZ, 85287, USA. [3]Institute of Physical Chemistry and Abbe Center of Photonics, Friedrich-Schiller-University Jena, Helmholtzweg 4, 07743 Jena, Germany. ✉e-mail: benedictdied@gmail.com

reconstruction unmixes several overlapping down-modulated object components and places each of them at the correct location and phase in the Fourier transformed final image. Coherent illumination has the great advantage of providing near 100% pattern contrast up to the highest frequencies supported by the illumination system. Although SIM has the same theoretical passband limit as a confocal microscope[9], the signal-to-noise ratio, especially at high spatial frequencies, is better due to the very high contrast of the high frequency illumination. Coherent illumination is therefore a key component of SIM systems that aim for high resolution rather than optical sectioning.

Since its introduction[7,8,10], many have refined SIM, e.g. by replacing the illumination patterns of the classical two-beam interference and the resulting sinusoidal illumination in the sample plane with two-dimensional[11], speckle-like[12] or hexagonal illumination patterns[13], in order to simplify the setup or to achieve faster data acquisition[14]. Other efforts optimized the image processing algorithms e.g. using deep learning[15] or GPU-assisted reconstruction algorithms to realize real-time SIM[16–19]. Recently integrated solutions realize SIM illumination by means of photonic chips[20], metamaterials[21] or optoacoustic Bragg cells[22]. There are several ways to implement even classical two-beam SIM, such as using a Michelson interferometer[23] or a fiber array[24]. Besides the use of a classical diffraction grating in a conjugate image plane, the use of a spatial light modulator allows a flexible and accurate, non-mechanical manipulation of the phases and diffraction orders. Liquid crystal on CMOS (LCoS) spatial light modulators (SLM's)[25,26] represent relatively straightforward to implement technologies, but are expensive and slow. Alternative approaches, such as the recently introduced method using galvo scanners[27,28] to manipulate the phase and rotation in the BFP, require a high effort in aligning the optical components. Digital micromirror devices (DMD's) on the other hand are a relatively cheap and fast alternative[29,30], but are amplitude rather than phase modulators, so they are less efficient in some configurations. However, DMD-SIM systems are more challenging to design and align due to complex diffraction effects caused by the tilt of the DMD micromirrors[31–33].

Those who do not have the necessary budget of several hundred thousand euros for the procurement of a commercial grade SIM instrument are therefore often confronted with high complexity in the realization of a SIM microscope. Since for SIM illumination, the optical path can be fairly complex, the possibility of building a super-resolution microscope is limited for non-specialists in optics. Established biological research groups can often draw from a selection of microscopes in their department or the local core facility, though few can keep up with current development in modern microscopy methods, which often means that better resolution requires the purchase of a new microscope and a corresponding budget.

The field of "Open Microscopy"[34] tries to solve this dilemma by providing detailed blueprints for replication verified by peer review and by using frugal science to keep the price low. With our recently introduced SIM module for the UC2 system[35,36], we have presented a monolithic module that complements the cube-based 3D printed microscope with structured illumination imaging capability. However, the system is aimed more at education with a relatively small DMD display. In this work, we build on these principles to design an open-source SIM add-on for commercial microscope stands, which is suitable for addressing scientific questions.

Our work is designed to complement significant advancements in the field, such as the OpenSIM modules discussed in ref. 26, by leveraging coherent illumination to achieve superior modulation contrast. Unlike the multi-color-based illumination of OpenSIM, which enhances existing commercial microscopes and relies on incoherent LED (light emitting diodes) illumination, our method not only offers improved image quality but also integrates open-source software. This approach makes it more accessible and cost-effective, addressing common concerns with proprietary systems and high costs associated with other SIM technologies. Similarly, with our recently introduced (coherent) mcSIM framework[32], we present a fully open-source system that includes construction plans, CAD tools and control/reconstruction. However, it requires a large budget and optical knowledge to replicate the setup, as high-class optical components are used

and the setup of a complete microscope on the optical table is necessary, making replication for less-skilled people difficult.

To foster inexpensive access to SIM imaging beyond previous approaches, we present a fully open-sourced, dual-color (488/635nm) structured illumination microscopy add-on module that can be easily adapted to commercially available fluorescence microscopes. The add-on includes both hardware and software components and is supported by extensive documentation (including parts lists and setup videos). On the hardware side, we present an open-source illumination engine based around a DMD. With this system, we achieve super-resolution enhancements of up to 1.5 times to the Abbe limit. This resolution enhancement could be further increased by using more aggressive SIM patterns.

We provide a SIM acquisition and reconstruction plugin for the Python-based open-source microscopy control software ImSwitch[37], enabling live stream SIM image observation and further seamless integration in automated microscopy workflows.

Our add-on is based on low-cost components, which greatly reduces the overall cost so that the final build can be done for around three thousand euros. This, together with the reduced complexity of the build process, increases the reproducibility of our approach. The concept of open-source makes this project accessible to all in need of advanced imaging systems, rendering this cutting-edge technology affordable for a broad range of research laboratories.

## Methods
### Multicolor DMD alignment
In this work (see rendering in Fig. 1, we rely on a DMD to generate structured illumination patterns due its low cost, large spectral acceptance range, fast speed, and reproducible quality. As discussed above, DMD's are beam shaping devices which are commonly used in commercial video projectors and have recently been applied to a wide variety of biomedical applications, including SIM[31–33,38,39]. However, DMD's are primarily designed for use with incoherent light, and working with coherent light introduces several challenges. Most significantly, the complex DMD diffraction hampers their use in multicolor laser systems as compared with LCoS SLM's. Additionally, DMD's can introduce aberrations into the illumination system both due to the mirror arrays not being optically flat and also the presence of a thick glass window which is a few millimeters away from the vacuum-sealed micromirrors.

A DMD consists of a rectangular array of micromirror "pixels" arranged on a square lattice in the DMD backplane. Each mirror is mounted on a micromechanical swivel which is controlled by electrodes that can dynamically position the mirror in one of two orientations, which we refer to as the " + " or " − " orientation. The mirrors can be independently controlled to display an arbitrary binary pattern. Brightness in video-display applications is typically controlled by temporal multiplexing. Due to the tilt of the micromirrors, DMD's are essentially blazed diffraction gratings, which can have high diffraction efficiency into a non-zero diffraction order, but usually cannot be optimized for use with the zeroth order.

A coherent plane wave incident on a DMD with mirrors all in the + state will be diffracted into many different orders (Fig. 2A) due to the periodic mirror structure. The relative intensity diffracted into the various orders is modulated by an envelope function, the blaze envelope, which has its maximum at the direction of specular reflection from the mirrors. Achieving efficient diffraction requires one of the diffraction orders to be aligned with the center of the blaze envelope, in which case we say the system satisfies the blaze condition. When the DMD displays a pattern, using mirrors in both the + and − states, the pattern structure generates additional subdiffraction orders about the main diffraction orders (Fig. 2B). If the primary diffraction order does not meet the blaze condition, the positive and negative suborders will have different intensities. This distortion is particularly deleterious in SIM because unequal suborder intensities result in reduced SIM modulation contrast, which rapidly degrades the recoverable superresolution information. We discuss this effect extensively in previous work, and find that an imbalance factor of $\eta \leq 1$ between the two

**Fig. 1 | Setup overview and selection of components. A** the complete openSIMMO (open-source structured illumination microscopy module) setup attached to a Nikon Ti2-A Eclipse. **B** The beam path of the two lasers is folded to minimize the footprint and meet the blaze condition. **C** Aside from the laser-cut housing and a minimal number of external optics components, the mechanical design relies on the RailOptics system that enables translation of optics along the optical axis. **D** the digital mirror device (DMD) module is positioned with a customized optomechanical mount and driven via the HDMI port of a Raspberry Pi. **E** off-the-shelf optics are adjusted using 3D printed adapters mounted on a laser-cut base plate.

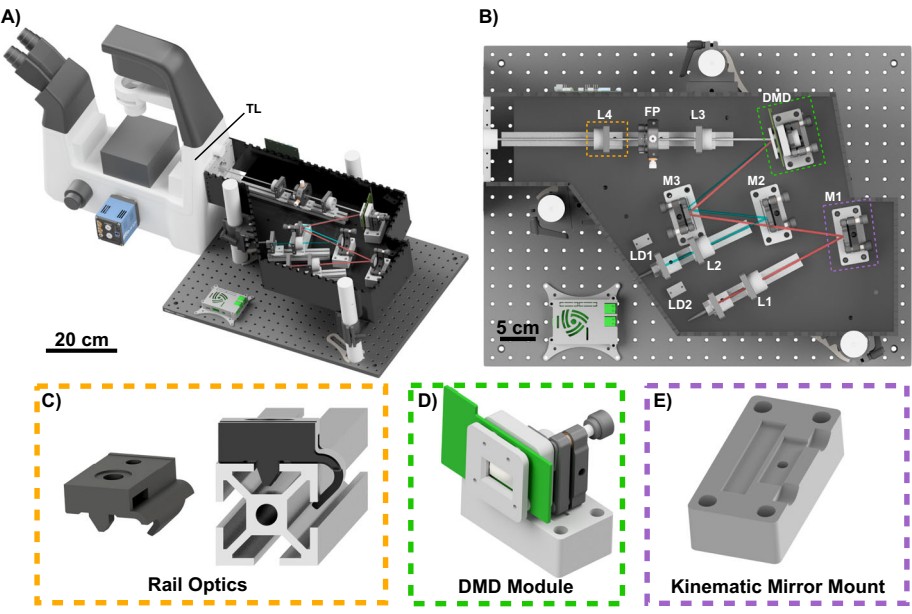

beams causes the SIM modulation contrast to be degraded to $m = 2\eta/(1 + \eta^2)$[32].

Multicolor DMD operation is particularly challenging because the output angle of the diffraction orders are wavelength dependent, but the blaze envelope direction is not. As such, multicolor operation requires either carefully tuning the illumination wavelength[40], using different input directions for different wavelengths[32], or accepting reduced SIM modulation contrast for certain wavelengths[39]. This problem is unique to DMD's, as LCoS SLM's are usually operated using the zeroth diffraction order, which does not exhibit chromatic dispersion.

Recent work has surmounted these challenges to realize multicolor coherent DMD-SIM using up to 3-colors[32,40]. This work fully characterized the possible multicolor solutions which satisfy the blaze condition, but using a model specialized to DMD's with corner illumination pixels (CIP's). CIP's have a rotation axis at a 45° angle to the mirror lattice but in the same plane, and rotate symmetrically about this axis to orient the mirror normals ±12° away from the DMD backplane normal to reach their + and − states respectively (Supplementary Note S7C). However, in this work we use the widely-available and cost-effective DMD evaluation module DLP4710EVM-G2 (Texas Instruments, USA, TX) which features tilt and roll pixels (TRP's)[41]. TRP's use a composite rotation which effectively rotates the mirrors ~17° about the $x$ and $y$ mirror lattice axes in the + and − orientations respectively (Supplementary Note S7D).

To rationally design our multicolor coherent DMD-SIM module, we extend the CIP diffraction model of ref. 32 to DMD pixels with an arbitrary rotation axis, which include TRP's as a special case. We leverage this model to select a geometry which enables 2-color SIM using 488 nm and 635 nm excitation.

To develop this model, we suppose our DMD mirror lattice has pitch $d$ and the mirrors rotate about an arbitrary axis $\hat{\mathbf{m}}^\pm$ through angle $\gamma^\pm$ as described by a rotation matrix $R^\pm = R(\hat{\mathbf{m}}^\pm, \gamma^\pm)$ to reach the ± orientations (Supplementary Note S7B). We then define a coordinate system with unit vectors $\hat{\mathbf{e}}_x$ and $\hat{\mathbf{e}}_y$ along the principal axes of the mirror lattice in the backplane of the DMD chip and $\hat{\mathbf{e}}_z$ normal to this plane and pointing away from the DMD. The DMD is illuminated by a coherent plane wave with incident direction described by unit vector $\hat{\mathbf{a}}$, and we will consider light diffracted into output direction unit vector $\hat{\mathbf{b}}$ (Fig. 2B). By construction, $a_z$ and $b_z$ are along the $\mp\hat{\mathbf{e}}_z$ directions respectively,

$$a_z = -\sqrt{1 - a_x^2 - a_y^2} \tag{1}$$

$$b_z = \sqrt{1 - b_x^2 - b_y^2}. \tag{2}$$

The incoming and outgoing directions must satisfy the diffraction condition of the mirror lattice,

$$b_x - a_x = n_x \frac{\lambda}{d} \tag{3}$$

$$b_y - a_y = n_y \frac{\lambda}{d}, \tag{4}$$

where $n_x$ and $n_y$ are integers indexing the diffraction orders.

To achieve high diffraction efficiency, the incoming and outgoing directions should also satisfy the blaze condition. To simplify our analysis of this condition, we introduce a new coordinate system, defined by unit vectors $\hat{\mathbf{e}}_i = R(\hat{\mathbf{m}}, \gamma)\hat{\mathbf{e}}_\alpha$ for $i = 1, 2, 3$ and $\alpha = x, y, z$. Since $\hat{\mathbf{e}}_3$ points along the micromirror normal, the blaze condition in these coordinates is

$$\begin{aligned} b_1 - a_1 &= 0 \\ b_2 - a_2 &= 0 \\ b_3 + a_3 &= 0. \end{aligned} \tag{5}$$

Combining the blaze and diffraction conditions, we have six unknowns, the components of $\hat{\mathbf{a}}$ and $\hat{\mathbf{b}}$, and seven equations. The system is over-determined and does not have any exact solutions in the general case. However, for specific choices of the mirror rotation matrix the above system of equations can have an exact solution. To understand when exact solutions exist, we rewrite eqs. (3) and (4) in the mirror basis and substitute eqs. (5) to find

$$\begin{aligned} b_3 &= \frac{n_x}{2R_{13}} \frac{\lambda}{d} \\ b_3 &= \frac{n_y}{2R_{23}} \frac{\lambda}{d}, \end{aligned} \tag{6}$$

where $R_{ij}$ are the matrix elements of the mirror rotation matrix. Eqs. (6) can only be satisfied if the ratio of $R_{13}$ and $R_{23}$ is a rational number. For example, for CIP's $R_{13} = -R_{23}$ implying exact solutions exist for diffraction orders $n_x = -n_y$ (Supplementary Note S7C).

In the general case, we can alternatively view this as an optimization problem where we regard the diffraction equations as constraints and the

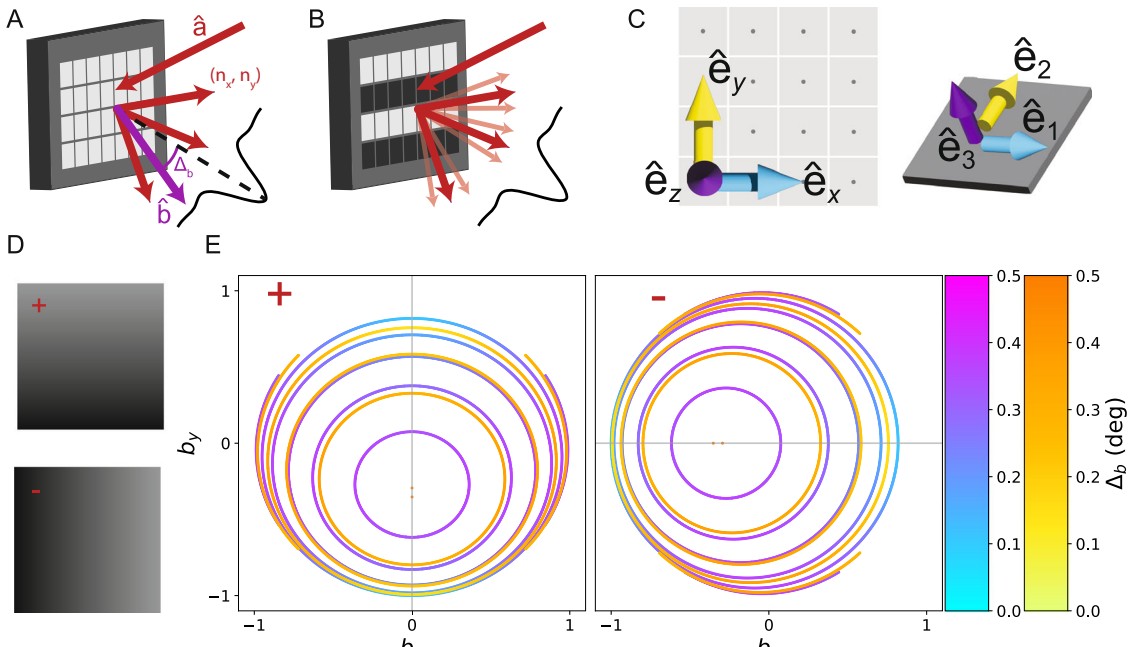

**Fig. 2 | Simulation of digital mirror device (DMD) as a spatial light modulator (SLM) with coherent illumination. A** Coherent light incident on the digital mirror device (DMD) from direction $\hat{\mathbf{a}}$ is diffracted into orders $(n_x, n_y)$ whose strengths are modulated by the blaze envelope (black curve). We determine which output directions $\hat{\mathbf{b}}$ (purple) are allowed (red), and how much they deviate from the center of the blaze envelope, $\Delta_b$. **B** When a pattern is displayed on the DMD, the additional structure produces subdiffraction (light red) centered around the primary diffraction orders (red). **C** We define one basis aligned with the DMD backplane $\hat{\mathbf{e}}_x, \hat{\mathbf{e}}_y, \hat{\mathbf{e}}_z$ and a second basis which is adapted to the tilted mirror $\hat{\mathbf{e}}_1, \hat{\mathbf{e}}_2, \hat{\mathbf{e}}_3$. The two bases are related by a mirror-state dependent rotation matrix $R^{\pm}(\hat{\mathbf{m}}, \gamma)$. **D** tilt and roll pixels (TRP) support two states, "+" and "-" (left). The mirror rotation axis nearly

coincides with $\hat{\mathbf{e}}_{x/y}$ for the $\pm$ states, respectively. The darker corner of the micro-mirror is the landed side. **E** For each given diffraction order, many different input/output direction pairs lead to optimal solutions. We display these as curves in output direction space, parameterized by unit vector components $b_{x/y}$. We determine the most nearly blazed diffraction solutions for 488 nm (blue to purple) and 635 nm (yellow to orange) incident light for several different diffraction orders for both the $\pm$ (left/right). For the + mirrors, we consider diffraction orders $(0, n)$. For 488 nm, the innermost curve represents the $(0, -6)$ order, and increasing area corresponds to lower order. For 635 nm, the first curve corresponds to $(0, -4)$. The - mirrors show identical structure under a 90° rotation.

blaze condition violation as a function we would like to minimize. To this end, we define a cost function

$$\mathcal{C}(\hat{\mathbf{a}}, \hat{\mathbf{b}}) = (b_1 - a_1)^2 + (b_2 - a_2)^2, \quad (7)$$

which characterizes how much input-output direction pair $\hat{\mathbf{a}}$ and $\hat{\mathbf{b}}$ violate the blaze condition. This cost function is not unique, as we could include additional terms penalizing deviations in $b_3 + a_3$. But as we will see, this choice is expedient because it is easy to solve, and it reproduces the exact solutions if they exist.

We minimize the cost function subject to our constraints using the method of Lagrange multipliers, which supplies us with a new set of equations to replace the blaze condition,

$$\nabla \left[ \alpha \left( b_x - a_x - n_x \frac{\lambda}{d} \right) + \beta \left( b_y - a_y - n_y \frac{\lambda}{d} \right) + \mathcal{C}(\hat{\mathbf{a}}, \hat{\mathbf{b}}) \right] = 0, \quad (8)$$

Where $\alpha$ and $\beta$ are Lagrange multipliers, and the gradient is taken with respect to $a_x, a_y, b_x$, and $b_y$. The details of solving this system are provided in (Supplementary Note S7A).

For TRP's in the − state, if we divide equations (6) we find $R_{13}^-/R_{23}^- = 90.523$ (Supplementary Note S7D), where $R^-$ is the rotation matrix describing the mirror orientation for the - state mirrors. This implies there are no exact solutions, because even if the ratio is rational, the large diffraction orders required to satisfy it are not physical. We can search for approximate solutions for diffraction orders where this ratio is large. In

particular,

$$
\begin{aligned}
n_x &\leq 0 \\
n_y &= 0 \\
b_3 &= \frac{n_x}{2R_{23}^-} \frac{\lambda}{d}.
\end{aligned}
\quad (9)
$$

These solutions would be exact, except for the small component of the rotation axis along the $\hat{\mathbf{e}}_z$ direction. The situation for the + state is identical under 90° rotational symmetry.

We solve this system of equations for the TRP pixels with both 488 nm and 635 nm to determine the optimal input and output directions and find two-color compromise solutions (Fig. 2E). Since each choice of mirror orientation and diffraction order produces a different solution, we solve for both the + and − orientations and a sequence of low diffraction orders matching the form determined in eq. (9), in particular $(n_x, n_y) = (n, 0)$ for the − mirrors, and $(0, n)$ for the + mirrors, where $n = -1, \dots, -6$. For most diffraction orders, we find that the solution directions are not unique, but form a one-parameter family. We display these solutions as curves in output unit vector space, parameterized by $b_x$ and $b_y$ (Fig. 2E). The corresponding input directions for any point on these curves can be determined from the diffraction conditions, eqs. (3) and (4). For 488 nm illumination, the innermost closed curve corresponds to the $(0, -6)$ order, and the larger curves correspond to progressively decreasing orders. For the 635 nm illumination, the solution for the $(0, -6)$ and $(0, -5)$ orders are single points, and the innermost closed curve corresponds to the $(0, -4)$ order. The larger curves again correspond to progressively decreasing diffraction orders.

While the solutions displayed in Fig. 2E) are all optimal in the sense of eq. (7), they do not have identical blaze condition violation angles, $\Delta_b$

because eq. (7) does not consider the $\hat{\mathbf{e}}_3$ direction. Furthermore, solutions for different diffraction orders may have different blaze angle violations. However, for the TRP pixels we find the blaze condition violations angles are typically ≤0.5°, and the magnitude is shown using the color scale in Fig. 2E.

From the simulations, we conclude that the TRP - mirror state supports nearly-blazed 2-color solutions for the 488 nm ($-5, 0$) order and the 635 nm ($-4, 0$) order which nearly overlap due to the fact 635/488 ≈ 5/4. While there are many possible choices of input/output directions, for experimental convenience, it is convenient to work with directions where the beams stay in the same 2D plane. For the TRP - state, this corresponds to light incident in the $xz$ plane. Focusing on this plane, we have a solution with output direction that forms an angle of ≈ 20° with the DMD backplane normal. For 488 nm the optimal solution is incident at an angle of 55.9° and diffracts at an angle of 22.1° with a blaze angle violation of 0.29°. For 635 nm the optimal solution is incident at an angle of 52.9° and diffracts at an angle of 19.1° with a blaze angle violation of 0.3°. The input angle of one or both beams must be adjusted to ensure the output angles are the same, entailing some additional blaze angle violation. For example, if the 635 nm beam is aligned to the 488 nm beam, the incidence angles must be modified to 57.9°, and the blaze condition violation increases to 1.97°. In this case, using the 1.55 × patterns discussed in Supplementary Note S4 will lead to $\eta = 0.95$ and 0.86 for the 488 nm and 635 nm wavelengths respectively. In either case, the effect on the modulation contrast is small, resulting in $m \geq 0.99$.

While the previous solution achieves small blaze condition violation, it requires the DMD backplane to be tilted significantly with respect to the optical axis, which can introduce unwanted effects including optical aberrations and pattern focal plane variation across the field of view. To address these issues, we also consider solutions for the 488 nm ($-6, 0$) order and the 635 nm ($-5, 0$) order. In the optimal alignment for the 488 nm ($-6, 0$) order, the incoming beam is incident at 38.1° with blaze violation of 0.34° and DMD backplane tilt of 4.3°. Aligning the 635 nm order to the same output direction as the 488 nm beam results in an incidence angle of 41.6° with a blaze violation of 3.4°. In this case, using the 1.55 × patterns leads to $\eta = 0.94$ and 0.71, leading to modulation contrasts of 0.999 and 0.94 for 488 nm and 635 nm respectively. We choose to use this solution in the experiment.

Previous experiments have reported DMD mirror rotation parameters which differ somewhat from the nominal values[16,31,32]. To address this possibility, we infer the true mirror rotation parameters for the—state of our DMD by displaying a series of structured patterns on the DMD, recording the intensity of their various diffraction sub-orders on a camera, and fitting our model to the results[32] (Supplementary Note S7E). We determine that the mirror rotation axis is ~13° different than the nominal value, and the mirror rotation angle is 17.7°. These differences imply typical optical blaze condition violations are on the order of 3°, somewhat larger than the values obtained for the nominal DMD parameters we considered above. Using these values, we estimate $\eta = 0.64$ and 0.67 and maximum achievable modulation contrasts to be 0.91 and 0.93 for 488 nm and 635 nm respectively.

We have previously demonstrated DMD-SIM systems can achieve high-quality three color imaging[32] by exploiting both the + and − mirror states. A similar approach is possible with openSIMMO, although it is somewhat more complicated due to the TRP geometry which would require working with out-of-plane optics to address the other mirror state.

## Mechanical and Optical Design of the SIM Setup

Our goal is a compact, low-cost setup that adapts with a large variety of different microscopy bodies. The developed hardware represents a compromise between price, availability, complexity and achievable resolution enhancement. A laser-tight custom-designed DIY enclosure ensures that lenses, mirrors, and SLM have predetermined positions, streamlining the assembly process. This reduces the alignment effort to a minimum, with assembly by semi-experienced person trained in optics being possible in about 2 hours (provided that all parts are prepared in advance), based on detailed step-by-step documentation[42]. Using different adapter plates (e.g.

Thorlabs SM1A30 for the Nikon Ti2) and tube lenses (e.g. Thorlabs AC254-200-A-ML, $f\prime = 200\,mm$ for the Nikon Ti2) respectively enables the adaptation to different microscopy bodies. To demonstrate the openness of our setup, we provide an adapter for the OpenFrame project[43], promoting open science by enhancing an existing system with an imaging modality instead of developing a new one. This increases the likelihood of implementation due to the widespread use of OpenFrames (see Fig. S8). A complete bill of materials can be found in Sec. S10).

For the light source, we employ two single-mode fiber-coupled (core 4 $\mu m$, non-polarization maintaining) diode lasers (openUC2, Jena, Germany) with operating wavelengths of 488 nm and 635 nm. Their output power is tuned via PWM signals from an ESP32 microcontroller (binary pulse width modulated, PWM, signal at $f_{\text{duty cycle}} = 5$ kHz at $10Bit$ resolution).

The optical setup geometry, as illustrated in Fig. 1, was designed with the help of the previously calculated optimal beam incidence angles for the 488 nm ($-6, 0$) and the 635 nm ($-5, 0$) orders using the − mirrors. The laser light emerging from the single-mode fibers is collimated using achromatic lenses of 50 mm focal length (AC-254-50-A M, Thorlabs). Then the two beams are combined and directed towards the DMD chip using mirrors (PF10-03-P01, Thorlabs), which are arranged to minimize the overall footprint of the assembly. The DMD displays the desired SIM patterns, and we direct light diffracted from the − mirrors towards the collection optics.

After the DMD, the illumination pattern is relayed by a 1:1 telescope consisting of two 75 mm focal length lenses (AC-254-75-A M, Thorlabs) and a Fourier mask is deployed to intercept the Fourier plane and obstruct the main diffraction order, leaving only the two subdiffraction orders for 2D SIM illumination. A template for a Fourier mask was first drawn with AutoCAD and printed out on paper. The mask shape was cut into a 1-inch radial shape using black aluminum foil (Thorlabs BKF12). The foil was then glued to the printed template and pierced through the foil with a needle according to the printed template. A pizza polarizer, as proposed in[25], was not included because the increased contrast advantage was not outweighed by the increased cost and complexity of construction (i.e. due to the compact telescope, the small diameter of the Fourier plane makes it difficult to arrange individual quarter-wave plates). A tube lens (Thorlabs AC-254-200 A M) then focuses the sub-order beams on to the pupil plane of the microscope objective (60x/1.4 NA, Nikon, Japan), which is part of an inverted microscope stand (Ti2-A Nikon, Japan).

The patterned light excites fluorescence from the sample, which is recaptured by the objective lens, passes through a dichromatic beamsplitter (N-SIM 405/488/561/640, Nikon, Japan), and is focused onto the camera sensor (PCO Edge 4.2, Kehlheim, Germany) with a pixel pitch of 6.5 $\mu m$ by a tube lens inside the microscope body. The effective pixel size accounting for this camera and the 60× objective was 6.5/60 = 108 nm (5.8/60 = 97 nm for the Daheng camera), except in the dual color image and the test experiments at different gratings (Fig. S3) where we used an additional post-magnification of 1.5 yielding an effective pixel size of 6.5/60/1.5 = 72.2 nm. If we compare this with Nyquist's sample criterion, for the emitted light of 510 nm and 665 nm, we obtain 510 nm/(4 × 1.4) = 91 nm and 655 nm/(4 × 1.4) = 117 nm maximal sampling distance respectively. We therefore slightly undersampled the green channel of Figs. S1 and S2, whereas Figs. 3, 4, S3 and S4 were sampled correctly. The slight undersampling did not result in visible artifacts, presumably since the SIM processing at these frequencies strongly emphasizes the first order signal which stem from a non-aliased region with good signal to noise ratio. Note that the reconstruction algorithm in SIM typically leads to reconstructions at half the pixel size of the measurement to account for the extra resolution arising during SIM reconstruction.

Alternatively, for some experiments, we used an industry-grade USB3 camera (Daheng, MER2-230-168U3M, China) to show the ability of a cost-effective CMOS camera applied to scientific imaging. The Daheng camera uses an IMX174 sensor which has a pixel pitch of 5.8 $\mu m$ and an effective pixel size of 97 nm. It matches the Nyquist sampling criterion for 488 nm excitation with 60x microscope objective. In order to demonstrate the advantages of SIM in optical sections with a large field of view, we also used a

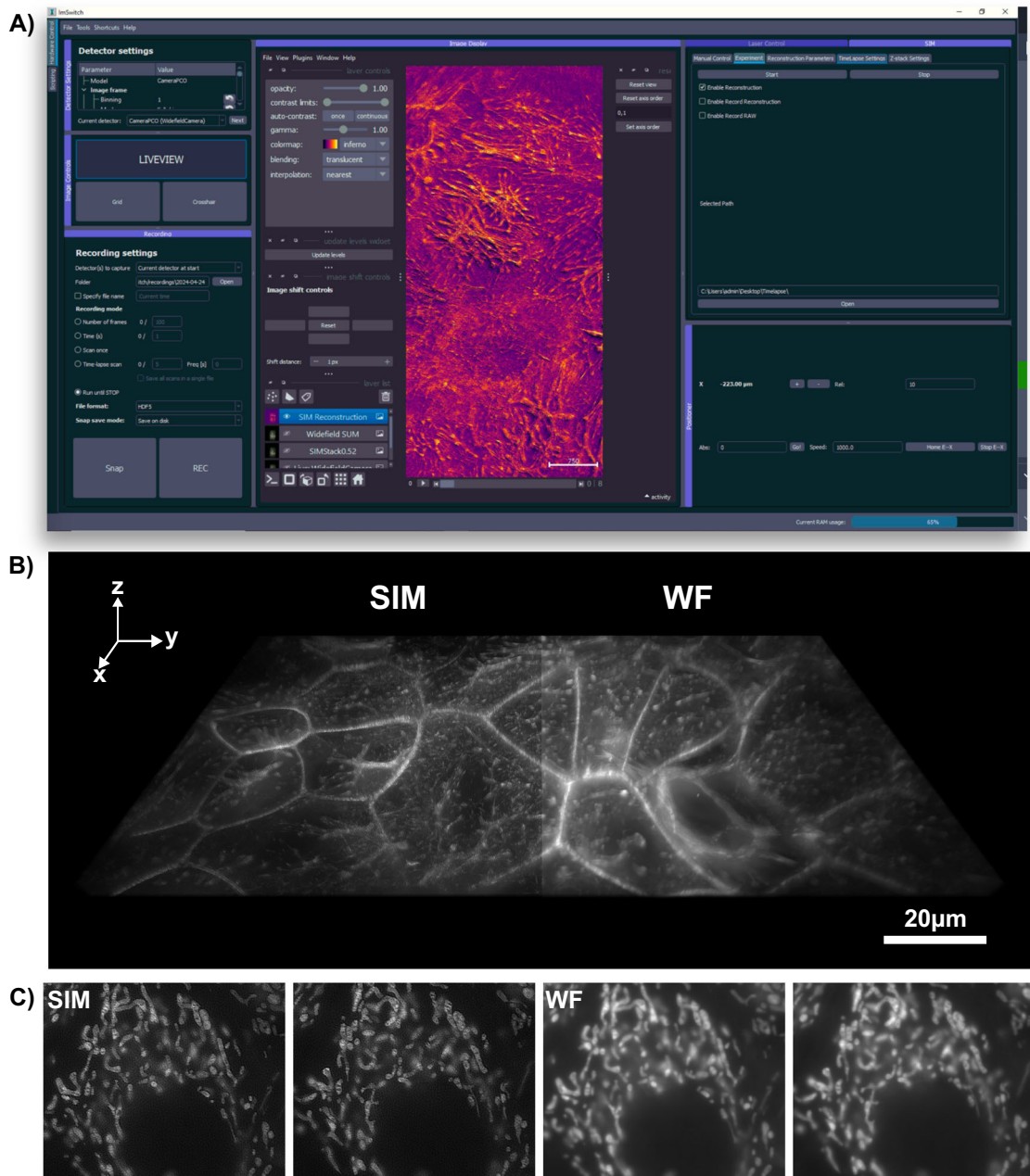

**Fig. 3 | Results of the structured illumination microscopy (SIM) module attached to an off-the-shelf microscopy body. A** Integration of real-time SIM reconstruction into the open-source microscopy control, acquisition, and processing software, ImSwitch. The installable SIM plugin allows users to input additional experimental parameters, managing pattern display and camera frame acquisition across different manufacturers. The resulting images are displayed as layers in napari and can directly be stored to the disk. **B** 3D stacks and **C** time-lapse data, exemplified here with Mitotracker-labeled HeLa cells. The enhanced optical sectioning compared to pseudo widefield (WF) images is clearly evident.

lower magnification objective (20x/0.75 NA, Nikon, Japan) in addition to the super-resolution approach.

Our mechanical design integrates commercial parts, craft materials, and 3D parts to create a stable, adjustable, and affordable experimental platform. Specifically, we incorporate two principal types of mounting hardware. First, aluminium profiles — referred to as "RailOptics" — afford critical adjustability for components along the optical path, such as the laser collimation lenses and the telescope. These are attached to a 3D-printed part (see Fig. 1B–E) that clips onto the aluminium profile, allowing for precision movement along the optical axis and facilitating quick changes and adjustments. Second, specialized 3D-printed bases house kinematic mirror

mounts (KM100, Thorlabs, US), provide fine angular adjustments to ensure the mirrors and the DMD chip are precisely aligned with the optical axis.

## Automation of the Image Acquisition and Reconstruction
OpenSIMMO (open-source SIM Module) is designed to achieve high-resolution SIM reconstructions in real-time during in vitro experiments. The hardware control, image acquisition, and data reconstruction must be orchestrated so that the high-resolution data can be displayed while live cell experiments are conducted. The open-source control platform ImSwitch (Fig. 3A) controls the various hardware components and synchronizes the individual modules. A customized plugin extends this napari-centric[44],

**Fig. 4 | Comparing widefiled and structured illumination microscopy (SIM) images of multi-color excitation. A** SIM Reconstruction and pseudo widefield comparison of Actin (488nm) and Mitochondria (635nm). The sample is GATTA-Cells 4C with stained huFIB cells (F36924, ThermoFisher Scientific, Germany). The structured illumination microscopy (SIM) reconstruction clearly reduces the background and improves the resolution depicted in the line profile **E. B** The comparison shows how SIM recovers the filamentous structure of actin in AF488 (Phalloidin) labeled samples zoomed in **C/D**. The sample is FluoCells prepared slide with stained BPAE cells (F36924, ThermoFisher Scientific, Germany). **E** Line plot showing the pseudo widefield image and SIM reconstruction of mitochondria from A along the magenta line. **F** Line plot showing the actin filaments from B along the magenta line. The SIM reconstruction separates previously unresolved filament structures.

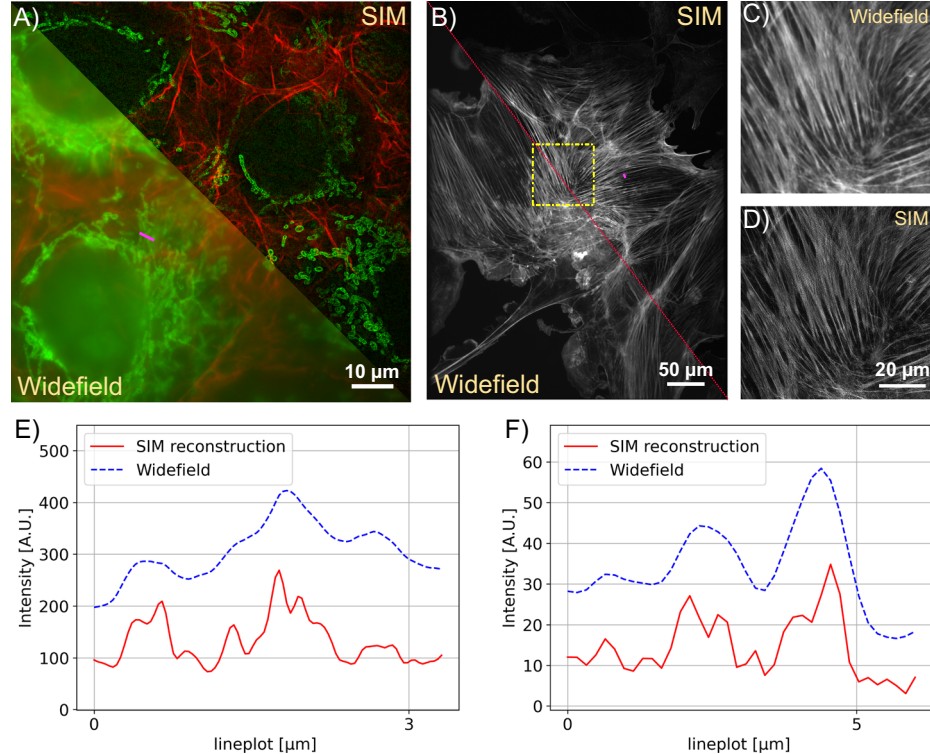

Python-based software with the option of entering SIM parameters and starting and controlling various experiments.

Here, openSIMMO benefits from a variety of different open-source algorithms for reconstructing the raw SIM images into a super-resolution result. napari-SIM[45,46] supports reconstructing raw data stacks in real-time after prior calibration of the illumination pattern parameters. The result is displayed directly in the napari viewer. The calculation can be accelerated if a graphics card is available. The same applies to the mcSIM library[47], which is a Python package that can also be integrated with napari. Alternatively, the stacks can be imported into ImageJ and reconstructed with fairSIM[16].

One difficulty is the lack of trigger input/output on the DMD evaluation board. An effective hardware-controlled and time-synchronous display of the previously calculated SIM patterns for the different lenses and wavelengths on the DMD was therefore not possible. For this reason, an additional control device, the Raspberry Pi 3b (Raspberry pi foundation, UK), was integrated into the workflow (Fig. 3). A REST-API-based display server streams the patterns stored on the device via HDMI to the DMD evaluation module and can simultaneously output trigger pulses for the camera. ImSwitch starts the continuous image acquisition via the "fastAPI" interface and sends the collected image stack containing images for all illumination patterns, wavelengths and focus positions to the reconstruction algorithms.

Compared to hardware trigger-based displays, the maximum possible recording speed is limited by the Raspberry Pi's refresh rate of 60 Hz. However, the limiting factor in the experiments shown here was the required minimum exposure time of ≥50 ms set by the relatively weak lasers.

To control additional hardware components, such as the focusing motor and incubation unit, the UC2-REST system was used, which is fully accessible via a USB serial connection through the ImSwitch GUI[48] (see Fig. 3). The ESP32-based system enables TTL-based control of the laser intensity, as well as the realization of a heating chamber to perform live cell experiments. A heating plate from a 3D printer (Prusa Mini, Prag, Czech Republic) was mounted in conjunction with a temperature sensor in a 3D printed case on the sample stage. A PID controller running on the microcontroller regulates the temperature to a constant value of 37 °C. To produce

image stacks in focus, a NEMA17 motor was attached to the manual focus drive of the microscope by means of a belt drive.

Our system can be easily adapted to a wide variety of different hardware due to the flexibility of ImSwitch. For example, ImSwitch offers a large variety of different camera drivers hardware adapters. In this work, work we have used this flexibility to profile openSIMMO using both scientific and industrial CMOS cameras.

### Reporting summary
Further information on research design is available in the Nature Portfolio Reporting Summary linked to this article.

### Results
A primary goal of openSIMMO is to make super-resolution microscopy using SIM available to a large user base. The associated need to develop an affordable, compact, easy-to-build system composed of widely available parts has limited the space of possible designs and led to a slight loss in maximum resolution, since we, for example, omitted the use of the pizza polarizer, which is required to achieve maximum modulation contrast especially for high-NA objective lenses. We characterize the system as follows, both in terms of biological and synthetic calibration samples, regarding the obtainable lateral and axial resolution, as well as the ability to use multiple colors sequentially and to record time-lapse series.

Here we worked with pattern periods of 310 nm for the 488 nm excitation and 425 nm for the 635 nm excitation (Supplementary Note S4). These patterns support increasing the maximum detectable spatial frequency by a factor of ~1.55, ignoring the effect of the Stoke's shift.

First, the maximum resolution of the setup was measured with a SIM ArgoLight slide (Argolight SIM v1, SLG-008, Argolight, France) using the 488 nm excitation and the data was reconstructed with FairSIM. The Wiener filter parameter used for the reconstruction is 0.05. The finest line pair that can be resolved with the deconvolved pseudo widefield image is 210 nm. After reconstructing the data, the line pairs with a distance of 120 nm can be clearly resolved. (see Fig. S1A, C). This corresponds to a 1.75-fold improvement. Since this measurement tests the two-point resolution of the

system, this resolution enhancement is not directly comparable with the 1.55-fold enhancement expected from the increased spatial frequency support above the Abbe diffraction limit. Two-point resolution measurements are sensitive to image contrast, in addition to increased spatial frequency support. This large resolution enhancement could also be affected by the undersampling with 488 nm laser excitation without using the in-built 1.5× intermediate magnification lens. For getting higher signal of the raw data, the magnification lens was removed to increase the photon number on each camera pixel.

To characterize the multi colour performance of the system with biological samples, we imaged a fixed huFIB cells sample with multiple fluorescence markers (GATTA-Cells 4C, Gattaquant, Germany) and the results are shown in Fig. 4A, E. Here, the mitochondria are labelled with Alexa Fluor 488 (LOT 2615834). and imaged with the 488 nm excitation, while the actin is labelled with Alexa Fluor 647 and imaged with the 635 nm excitation. In addition to the clear increase in resolution, the improved optical sectioning should be emphasized. The fluorescent background is suppressed using Gaussian zero-order suppression (default parameters for Napari-SIM and FairSIM were $\alpha = 0.5$, $\beta = 0.98$ and a = 0.99, FWHM = 1.2), leading to an improved imaging of the cell structure. The cell image resolution in the 488 nm and 635 nm channel are validated with the Fourier-ring correlation (FRC) method[49] which estimates a resolution of 130 nm vs. 220 nm and 174 nm vs. 260 nm for the SIM and pseudo widefield image, respectively. A 3D stack can be done with moving the employed NEMA17 motor which is attached on the fine focus knob. The stack was captured with 260 nm spacing over total range of 10.4 $\mu m$ as show in Fig. 3B.

Our open-source SIM module is the centerpiece of a low-cost, open-source live cell imaging platform. The open-source nature of ImSwitch enables us to implement additional features such as autofocus which are vital for live-cell imaging and time-lapse recordings. We demonstrate these functionalities on the recording of living cells, which is realized using a self-built incubator module. A PID controller implemented in UC2-ESP firmware[50] ensures a constant temperature of 37 °C, enabling affordable in vitro imaging of living cells.

We demonstrate our instrument's live cell imaging capabilities by performing a time-lapse series of HeLa cells cultured with MitoTracker Green dye (FM LOT 1829928). Fig. 3C shows two of the total 30 time points ($t_{\text{period}} = 2$ min, $t_{\text{exposure}} = 50$ ms), where moderate bleaching and a focus drift of the objective of about 20 $\mu m$/h was observed. The fine focus knob was turned mechanically by the motor before each image series acquisition to balance the focus drift. For the future it could be compensated using an autofocussing algorithm implemented in ImSwitch. Compared to the pseudo widefield, improved optical sectioning is observed in the SIM reconstruction (Fig. 4). The resolution of the time-lapse pseudo widefield and SIM data are estimated to be 226 nm and 158 nm by using FRC, which gives a resolution enhancement of 1.43× .

For the time-lapse and 3D stack experiments presented here, we used the 60 × objective in combination with the Edge 4.2 sCMOS camera without the intermediate magnification lens and the data is ~10% undersampled at the ~520 nm emission wavelength corresponding to the 488 nm excitation. Since the OTF rapidly falls off at high spatial frequencies, we did not observe aliasing effects in the raw or reconstructed data.

## Discussion

In this paper, we introduce an illumination add-on specifically designed for structured illumination microscopy attached to a standard commercial fluorescence microscope body. Our configuration is enclosed within a laser-cut Plexiglas box, where components are mounted onto the baseplate similar to commercial illumination add-ons. The add-on is fully open-source, including a Python-based framework that has access to a large variety of different cameras and hardware control elements using napari and ImSwitch. We demonstrate that our setup, which in its minimal form incorporates an industry-grade camera, off-the-shelf motors and micro-controllers, and a DMD-based SIM module, enables a powerful SIM

upgrade to existing microscopes for approximately €3000, democratizing access to super-resolution imaging capabilities. Additionally, our adaptable control software supports various hardware configurations, potentially revitalizing outdated microscopes into state-of-the-art super-resolution devices for live cell experiments, thus giving scrap-heap equipment a second life. Our implementation simplifies the integration of super-resolution imaging into fully automated workflows, where e.g. automated sample preparation can be part of a larger pipeline. A remote-controllable REST-API integrates triggering of super-resolution imaging and reconstruction in a unified way. Every part of this project has been open-sourced, allowing for collaborative contributions. A step-by-step demonstration guides users through preparing the components, building and aligning the hardware setup, and configuring the system for operation, ensuring accessibility and ease of adoption.

By minimizing the use of commercial opto-mechanical parts and by using 3D printed parts instead, we have increased the reproducibility. Through a series of rigorous experiments, we consistently obtained reliable results. Our finalized setup enhances resolution, surpassing that of a traditional widefield microscope by a theoretical factor of 1.55. This extension can be easily attached to microscope bodies with an epi-fluorescence port, increasing its versatility. The accompanying software is intended for technically skilled biologists, lab technicians and tinkerer. Although we aim to lower the barriers to wider participation, especially for users from low-income countries, a certain level of technical knowledge is still required for setup and operation.

While openSIMMO achieves excellent balance between low-cost and high-quality super-resolution imaging, some of the hardware compromises limit some aspects of the setup's performance. In order to get an acceptable result in the current configuration with the selected lasers of limited power, the exposure time for each frame needs to be longer than 50 ms, hence limiting its use for rapidly moving objects in live-cell imaging. Additionally, longer exposures mean the speed limit of the DMD has yet been reached. An upgrade to a high power laser holds the potential to further enhance the setup's operation speed, further pushing the boundaries of its performance. Improving polarization control has the potential to improve imaging contrast (further quantified in Sec. S6) and thus the maximum achievable resolution improvement. For example, the implementation of circular polarization, e.g. with quarter-wave plates (Thorlabs, WPQSM05-X approx. 300 euros), is a cost-effective solution for improving contrast if the budget permits. While adding a third excitation laser would be difficult due to both cost and space constraints—since additional beams would interfere with existing optomechanical components—it's possible to expand biological use cases by using alternative dyes that work within the current dual-excitation scheme. With careful selection of dyes, more than two fluorophores can be imaged, allowing for imaging across different emission spectral regimes. The current setup is optimized for two distinct excitation colors, but a wider variety of dyes could enable more flexible and diverse biological applications without needing a third excitation source.

Alternatively, the existing openSIMMO design could be made more affordable by adapting widely used projectors from DLP 3D printers. Such a design could provide a powerful and potentially lower cost alternative for pattern generation that offers the possibility of triggering the camera, as recent work has shown[51].

openSIMMO could potentially be extended to 3D-SIM by modifying the Fourier mask to allow the main diffraction order through, leading to 3-beam interference. However, this would also require adding additional dynamical polarization control elements to the system, considerably increasing the cost.

We anticipate that openSIMMO will democratize super-resolution fluorescence imaging and significantly expand the availability of high-contrast coherent SIM instruments in the life sciences. We believe broader dissemination of this technology will have major impacts on fluorescence live cell imaging and improve the quality and reproducibility of imaging experiments generally.

## Data availability
All datasets have been made available under https://doi.org/10.5281/zenodo.12638005. The interactive documentation of the system can be found in the GitHub pages system: https://opensimmo.github.io/.

## Code availability
The repository with all design and production files can be found under this link: https://github.com/openSIMMO/openSIMMO/. The code for the SIM Plugin for ImSwitch is available in the following repository: https://github.com/openUC2/imswitch-sim.

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

## Acknowledgements

We acknowledge funding from the Nexus GIF grant no. G-1566-143.13/2023. We thank the free state of Thuringia for supporting this project. PTB, JU, and DPS acknowledge funding from grant no. 2021-236170 from the Chan Zuckerberg Initiative DAF, an advised fund of Silicon Valley Community Foundation and from Scialog, Research Corporation for Science Advancement, and Frederick Gardner Cottrell Foundation grant no. 28041. We acknowledge Frank Garwe for his assistance in the grant application and Gerhard Holst for constructive feedback on the camera. We also thank Nikon for allowing us to use an instrument to which our setup was attached to.

## Author contributions

Conceptualization: B.D., H.W., P.B. Data curation: H.W., P.B. Funding acquisition: B.D., D.P.S. Investigation: B.D. Methodology: B.D., H.W., D.P.S., J.U., P.B., R.H. Project administration: B.D. Resources: B.D., D.P.S., R.H. Software: B.D., H.W., P.B. Supervision: B.D. Validation: B.D., H.W., P.B. Visualization: B.D., H.W., P.B. Writing - original draft: B.D., H.W., P.B., R.H. with input from all authors. Writing - review & editing: All authors.

## Funding

## Competing interests

B.D. is co-founder of the company openUC2 that builds and distributes open-source microscopes. Both B.D. and H.W. are shareholders of that company. All other authors declare no competing interests.
