## [Transparent Peer Review File · Communications Engineering]

Fully-Automated Multicolour Structured Illumination Module for Super-resolution Microscopy with two Excitation Colours

Corresponding Author: Dr Benedict Diederich

Version 0:

Reviewer comments:

Reviewer #1

(Remarks to the Author)

In "Fully-Automated Multicolour Structured Illumination Module for Super-resolution Microscopy", Wang et al. present a well worked out system for adding a digital mirror device based structured illumination microscopy addition for almost any microscope. Overall the paper is well written and clearly presents their developments as well as data collected by such a system attached to a standard commercial microscope stand.

Although by no means the first module to convert a standard microscope stand to a SIM system this paper significantly adds to the information available. In particular they carefully cover the specific difficulties of using a DMD for laser based patterned illumination due to its behavior as a blazed grating. Over all this paper is worthy of publication but I feel it would benefit of revision in a few areas.

Major issues:

1) Resolution enhancement:

There are significant discrepancies in expected resolutions and measured data. Particularly in the fact that the produced images sometimes have better resolution than expected, eg at the bottom of page 19 the authors give number showing the resolution at 488 nm much better than the theoretically calculated 1.55x but at 635 nm it is worse a little. Although the authors do test with different stripe patterns (see supplemental material) my reading of the manuscript is that these experiments were with a fixed stripe ratio and an expected resolution increase of 1.55.

One possible explanation given is that the 488 nm images are under-sampled, but surely this would reduce the possible resolution enhancement, not increase it.

2) Polarization control and improved SIM imaging.

The paper mentions the importance of correct linear polarization for high contrast structured illumination in the introduction and initial sections and then abruptly says they are not controlling the polarization in the system description mid way down page 14. Presumably their relatively large stripe width and resolution gain far below 2 mean that they can still achieve reasonable stripe contrast without this. However, I suspect that the resolution improvement could be dramatically improved if they had higher and more consistent contrast across the different stripe angles.

I think the discussion could also do with a discussion of the limitations of this approach.

1) Additional colors: The authors have managed to accommodate 2 colors but I suspect adding a third would become extremely challenging in terms of both optimizing the blaze parameters and getting space to launch the third laser at the correct angle.

2) Adding Polarization control: Adding linear polarization control would significantly improve contrast and hence reconstruction quality.

3) Extending the system to 3D SIM.

Minor issues.

"The main advantage of incoherent SIM systems such as the commercially available Zeiss Apotome is an improved sectioning capability similar to that of confocal microscopy."

This sentence doesn't really seem to fit, incoherent SIM is not mentioned elsewhere and although true, it doesn't seem relevant.

"Due to the tilt of the micromirrors, DMD's are essentially blazed diffraction gratings, which are designed to optimize diffraction efficiency into a non-zero diffraction order."

DMDs are not "designed to optimize" non zero order intensity. They are designed to be manufactured and enable rapid switching of incoherent light between two paths. The blazed grating issues arise as the authors are using a coherent beam.

"If the primary diffraction order does not meet the blaze condition, the relative strength of positive and negative suborders will be distorted" Yes, but the distortion in question is having different intensities, I think this should be made clear.

"Aligning the 635nm order" Missing space after "nm"

"The effective pixel size is 108 nm, implying slight under-sampling with regard to the Nyquist limit for the 488 nm excitation, and correct sampling for the 635 nm excitation, which would be 187 nm and 239 nm respectively."

The stated numbers are not the Nyquist limit but the resolutions.

"We characterize the system as follows, both in terms of biological and synthetic calibration samples, regarding the obtainable lateral and axial resolution, as well as the ability to use multiple colors sequentially and to record time-lapse series."

The authors may have characterized the axial resolution but nowhere do they present that data, either as a vertical section, or as numeric data.

"Compared to the widefield, a much better optical sectioning is observed in the SIM reconstruction (Fig. 4)."

This is not shown at all, the images are much higher contrast but to demonstrate that they have better optical sectioning they need to present a XZ section or similar.

"Since the OTF rapidly falls off at high spatial frequencies, we did not observe aliasing effects in the raw or reconstructed data."

I don't see what the authors expected to alias and how the rapid fall off of the OTF saved them from it?

"A step-by-step demonstration of preparing, building, aligning, and starting the extension setup enables widespread adoption."

I don't understand what "starting the extension setup" means in this context, probably just needs to be reworded.

"Our finalized setup enhances resolution, surpassing that of a traditional widefield microscope by factor of 1.55."

No, this is your theoretical extension the measurements demonstrated both more and less than this.

"This extension can be effortlessly affixed to microscope bodies with an epi-fluorescence port, enhancing its versatility. The accompanying software is accessible to users of all levels of expertise."

I think all levels of expertise is stretching the point a little. I think a more realistic assessment of the expertise required to both setup such a system and then for users to use it would definitely benefit the readers.

Conclusion:

Overall this is a good paper, and should be published with minor updates.

Reviewer #2

(Remarks to the Author)

The authors propose an open-source design for a 2-color structured illumination module that attaches to commercial and open source fluorescence microscopes, providing a 1.55 fold improvement for a moderate cost of 3k. The openSIMMO is automated using open-source software and assembly is well documented to allow others to build their own module. Although other SIM modules have been already been published, openSIMMO is unique because it integrates seamlessly with other existing open-source software and hardware, such as ImSwitch and openFrame. It also provides high illumination pattern contrast thanks to the use of coherent illumination, which results in increased resolution. For these reasons, I believe that this paper will be of great interest to others of the community and the wider field, and that many teams will choose to build one.

I believe the openSIMMO will be easy to reproduce thanks to the detailed online documentation provided by the authors, as well as their use of the git platform for users to ask for assistance if needed.

[Major Comment]

In this paper, the authors are sharing the design and characterization of a new open-source structured illumination module. They provide a thorough characterization of the performances of the SIM module, and detailed instructions on how to build it. It is very appreciable that openSIMMO integrates with other open-source software and hardware, which helps with the maintainability of the whole open-source ecosystem, as well as facilitates user experience.

[Minor comments]

****Title****

1. consider "2 color" instead of "multicolor" for added clarity

****Introduction****

2. [p1] "Structured illumination microscopy [...] only requires few photons"

Could you provide citation/data to support this claim (or adjust the claim)

3. [p1-2] "SIM achieves its high resolution by illuminating fluorescent samples with a series of very fine periodic patterns of very high contrast and detecting images at various pattern positions for computationally reconstruction"

suggested edit

4. [p4] "Unlike the multi-color incoherent illumination of OpenSIM"

suggested edit for clarity

5. [p4] "The add-on includes both hardware and software components and is supported by extensive documentation (including parts lists and setup videos)."

The documentation is indeed extensive. The authors clearly invested a lot of time and effort to make sure that their SIM module can be built by non-experts.

6. [p5] "We provide a SIM acquisition and reconstruction plugin for the Python-based open-source microscopy control software ImSwitch"

It is great that the authors chose to integrate their microscope control with existing open source software, which will increase the maintainability and ease of use.

****Methods****

7. [p13] "Previous experiments have reported DMD mirror rotation parameters which differ somewhat from the nominal values [16, 26, 27]. To address this possibility, we infer the true mirror rotation parameters for the – state of our DMD by displaying a series of structured patterns on the DMD, recording the intensity of their various diffraction sub-orders on a camera, and fitting our model to the results"

Given the manufacturing tolerance of the DMD, as well as the difference you observed, what do you expect in terms of impact on performance for openSIMMO?

8. [p13] "This reduces the alignment effort to a minimum, with assembly by semi-experienced opticians being possible in about 2 hours (provided that all parts are prepared in advance), based on detailed step-by-step documentation [37]."
Ref 37 points to the UC2 website. I did not find the step by step documentation for this SIM module there.

9. [p15] first mention of openSIMMO as a name for the SIM module developed by the authors. Could you explain what the acronym stands for? You could also consider introducing it earlier in the paper for clarity.

Results

10. [p18] "The finest line pair that can be resolved with the deconvolved widefield image is 210 nm."

Could you indicate which method was used to do the deconvolution. This is important in the context of quantifying the resolution.

11. [p18] "This corresponds to a 1.75-fold improvement compared to the Abbe diffraction limit, whereby a 1.55-fold improvement would have to be expected. The difference could be due to the undersampling with 488nm laser excitation without using the in-built 1.5x intermediate magnification lens. For getting higher signal of the raw data, the magnification lens was removed to increase the photon number on each camera pixel."

Could you clarify? When was the 1.5x magnification used/not used. For the wide field image? For the SIM image? As deconvolution and SIM reconstruction both depend a lot of the signal to noise, it is important for the reader to understand the exact experimental conditions.

I also struggle to follow the reasoning regarding the 1.75x vs 1.55x resolution improvement.

12. [Figure 4] Could you comment on the artifacts visible on the image and due to the SIM reconstruction and leftover illumination frequency. It might be interesting to experiment with the reconstruction parameters.

Discussion

13. Could you say a word about how the pattern spatial frequency would be adjusted depending on the nominal values of the objective used and how it is done in terms of software and hardware? This is important given that users might have objectives with different NA, magnification, etc.

14. Could you add a line about what would need to be changed to adapt to other fluorescence microscopes (ie they have different tube length focal distance and will need a difference mechanical attachment).

Online resources <https://opensimmo.github.io/docs/>

15. "The module provides an enhancement factor of 1.75x compared to general widefield microscopy."

Is it 1.55 or 1.75? this contradicts the value given in the paper

16. It is great that the authors included detailed instructions regarding laser safety.

The explanations are detailed, illustrated with an abundance of pictures and several videos.

17. "Aligning the Fourier-filtering telescope -> The CAD files will come soon (file an issue or write use if we forgot - sorry!)"

Could the files be added?

Reviewer #3

(Remarks to the Author)

Please see the review report.

Version 1:

Reviewer comments:

Reviewer #1

(Remarks to the Author)

The authors have clearly addressed all my points and I feel the manuscript should now be published.

(extremely) Minor point:

"A step-by-step demonstration of preparing, building, aligning, and initializing the hardware extension enables widespread adoption."

I think I must be totally missing the point here but even with the rewording I still don't understand what the "initializing the hardware extension" refers to. Do the authors mean that they have guides for how to build, align and then run the imaging system?

Reviewer #3

(Remarks to the Author)

The authors have addressed all the previous comments in their revision. I am happy with the current version and I have no further comments. The manuscript can now be accepted as is.

Response to reviewers

We thank the reviewers for their careful reading of our manuscript and their helpful critiques and suggestions. Below we provide detailed responses to the issues they raised and outline our changes to improve the manuscript. We also include a revised version of the manuscript with changes highlighted in red. All line numbers discussed in our response refer to the red-lined version of the manuscript.

Reviewer #1:

In “Fully-Automated Multicolour Structured Illumination Module for Super-resolution Microscopy”, Wang et al. present a well worked out system for adding a digital mirror device based structured illumination microscopy addition for almost any microscope. Overall the paper is well written and clearly presents their developments as well as data collected by such a system attached to a standard commercial microscope stand.

Although by no means the first module to convert a standard microscope stand to a SIM system this paper significantly adds to the information available. In particular they carefully cover the specific difficulties of using a DMD for laser based patterned illumination due to its behavior as a blazed grating. Over all this paper is worthy of publication but I feel it would benefit of revision in a few areas.

We thank the reviewer for the positive evaluation of our work.

Major issues:

1) Resolution enhancement: There are significant discrepancies in expected resolutions and measured data. Particularly in the fact that the produced images sometimes have better resolution than expected, eg at the bottom of page 19 the authors give number showing the resolution at 488 nm much better than the theoretically calculated 1.55x but at 635 nm it is worse a little. Although the authors do test with different stripe patterns (see supplemental material) my reading of the manuscript is that these experiments were with a fixed stripe ratio and an expected resolution increase of 1.55.

One possible explanation given is that the 488 nm images are under-sampled, but surely this would reduce the possible resolution enhancement, not increase it.

We thank the reviewer for the opportunity to clarify this important point. Part of the discrepancy is explained by the difference between the Abbe resolution enhancement due to the larger area of spatial frequency support expected in SIM and the two-point resolution calibration provided by the argoSIM slide for the 488nm excitation.

The expected 1.55 \times enhancement quoted in the text is the ratio of the maximum spatial frequency that can be detected in our SIM setup compared with the Abbe limit, $2NA/\lambda$. On the other hand, the argoSIM slide measurement tests the two-point resolution of the system, which is sensitive both to the maximum detectable spatial frequency of the system, and to the enhanced contrast which is obtained in SIM due to deconvolution. Therefore, we do not expect the two-point resolution measurement to be directly comparable to the Abbe frequency enhancement calculation.

We have also clarified this point in the text by editing lines 406–410 to read “Since this measurement tests the two-point resolution of the system, this resolution enhancement is not directly comparable with the 1.55-fold enhancement expected from the increased spatial frequency support above the Abbe diffraction limit. Two-point resolution measurements are sensitive to image contrast, in addition to increased spatial frequency support.”

In regards to the undersampling, this effect should reduce the measured resolution of the widefield image. However, its effect on the SIM reconstruction is more nuanced. Due to the frequency down-

mixing, the SIM images will still contain significant super-resolution information, although the absolute highest frequency information will still be lost.

2) Polarization control and improved SIM imaging. The paper mentions the importance of correct linear polarization for high contrast structured illumination in the introduction and initial sections and then abruptly says they are not controlling the polarization in the system description mid way down page 14. Presumably their relatively large stripe width and resolution gain far below 2 mean that they can still achieve reasonable stripe contrast without this. However, I suspect that the resolution improvement could be dramatically improved if they had higher and more consistent contrast across the different stripe angles.

We agree with the reviewer that adding polarization control would lead to better pattern contrast. We omitted polarization control with the aim of making this instrument as affordable as possible.

To more clearly address the trade-off we are making by omitting polarization control, we have added “In our experiments, we manually optimized the maximum available pattern contrast in the sample plane by twisting the fibers to select the polarization state.” to the section “Effect of Polarization on Pattern Contrast”

I think the discussion could also do with a discussion of the limitations of this approach.

1) Additional colors: The authors have managed to accommodate 2 colors but I suspect adding a third would become extremely challenging in terms of both optimizing the blaze parameters and getting space to launch the third laser at the correct angle.

While adding a third excitation laser would be difficult due to both cost and space constraints — since additional beams would interfere with existing optomechanical components — it is possible to expand biological use cases by using alternative dyes that work within the current dual-excitation scheme. With careful selection of dyes, significantly more than two fluorophores can be imaged, allowing for imaging across different emission spectral regimes. The current setup is optimized for two distinct excitation colors, but a wider variety of dyes could enable more flexible and diverse biological applications without needing a third excitation source.

In our previous work, “Multicolor structured illumination microscopy and quantitative control of polychromatic light with a digital micromirror device” (BOE 2021), we have demonstrated that it is often possible to extend DMD SIM to three wavelengths by taking advantage of both the “+” and “-” mirrors. A similar approach could be adopted here. We have added a brief discussion of this possibility to lines 281–284: “We have previously demonstrated DMD-SIM systems can achieve high-quality three color imaging by exploiting both the + and – mirror states. A similar approach is possible with openSIMMO, although it is somewhat more complicated due to the TRP geometry which would require working with out-of-plane optics to address the other mirror state.”

2) Adding Polarization control: Adding linear polarization control would significantly improve contrast and hence reconstruction quality.

We addressed this point in response to one of the referee’s previous comments.

3) Extending the system to 3D SIM.

We have added a discussion of changes necessary to extend openSIMMO to 3D SIM with 3-beam interference in lines 493–496, which now read “openSIMMO could potentially be extended to 3D-SIM by modifying the Fourier mask to allow the main diffraction order through, leading to 3-beam interference. However, this would also require adding additional dynamical polarization control elements to the system, significantly increasing the cost.”

Minor issues.

“The main advantage of incoherent SIM systems such as the commercially available Zeiss Apotome is an improved sectioning capability similar to that of confocal microscopy.” This sentence doesn’t really seem fit, incoherent SIM is not mentioned elsewhere and although true, it doesn’t seem relevant.

We thank the reviewer for this suggestion. We have removed this paragraph from the manuscript.

“Due to the tilt of the micromirrors, DMD’s are essentially blazed diffraction gratings, which are designed to optimize diffraction efficiency into a non-zero diffraction order.” DMDs are not “designed to optimize” non zero order intensity. They are designed to be manufactured and enable rapid switching of incoherent light between two paths. The blazed grating issues arise as the authors are using a coherent beam.

We agree with the reviewer’s point. We have now clarified this point by changing lines 154–155 to read “Due to the tilt of the micromirrors, DMD’s are essentially blazed diffraction gratings, which can have high diffraction efficiency into a non-zero diffraction order, but usually cannot be optimized for use with the zeroth order”.

“If the primary diffraction order does not meet the blaze condition, the relative strength of positive and negative suborders will be distorted” Yes, but the distortion in question is having different intensities, I think this should be made clear.

We agree with the referee’s clarification and have modified lines 164–167 to read “If the primary diffraction order does not meet the blaze condition, the positive and negative suborders will have different intensities. This distortion is particularly deleterious in SIM because unequal suborder intensities results in reduced SIM modulation contrast...”

“Aligning the 635nmorder” Missing space after “nm”

We have corrected this typo.

“The effective pixel size is 108 nm, implying slight under-sampling with regard to the Nyquist limit for the 488 nm excitation, and correct sampling for the 635 nm excitation, which would be 187 nm and 239 nm respectively.” The stated numbers are not the Nyquist limit but the resolutions.

Yes indeed, our previous statement was incorrect in several aspects. The corrected paragraph now reads:

”The effective pixel size accounting for this camera and the 60× objective was $6.5/60 = 108$ nm ($5.8/60 = 97$ nm for the Daheng camera), except in the dual color image (Fig. 4) and the test experiments at different gratings (Fig. S3) where we used an additional post-magnification of 1.5 yielding an effective pixel size of $6.5/60/1.5 = 72.2$ nm. If we compare this with Nyquist’s sample criterion, for the emitted light of 510 nm and 665 nm, we obtain $510 \text{ nm}/(4 \times 1.4) = 91$ nm and $655 \text{ nm}/(4 \times 1.4) = 117$ nm maximal sampling distance respectively. We therefore slightly undersampled the green channel of Figs. S1 and S2, whereas Figs. 3, 4, S3 and S4 were sampled correctly. The slight undersampling did not result in visible artifacts, presumably since the SIM processing at these frequencies strongly emphasizes the first order signal which stem from a non-aliased region with good signal to noise ratio. Note that the reconstruction algorithm in SIM typically leads to reconstructions at half the pixelsize of the measurement to account for the extra resolution arising during SIM reconstruction.”

“We characterize the system as follows, both in terms of biological and synthetic calibration samples, regarding the obtainable lateral and axial resolution, as well as the ability to use

multiple colors sequentially and to record time-lapse series.” The authors may have characterized the axial resolution but nowhere do they present that data, either as a vertical section, or as numeric data.

We thank the reviewer for this helpful suggestion. In order to improve the visualisation, we added a plot of 3D data with a cross section in the supplementary material.

“Compared to the widefield, a much better optical sectioning is observed in the SIM reconstruction (Fig. 4).” This is not shown at all, the images are much higher contrast but to demonstrate that they have better optical sectioning they need to present a XZ section or similar.

We thank the reviewer for this helpful suggestion. In order to improve the visualisation, we added a plot of 3D data with a cross section in the supplementary material.

“Since the OTF rapidly falls off at high spatial frequencies, we did not observe aliasing effects in the raw or reconstructed data.” I don’t see what the authors expected to alias and how the rapid fall off of the OTF saved them from it?

We assume that in an imaging system with undersampling, information coming from spatial frequencies which are finer than the pixel sampling will appear aliased. This means that this information appears as if it comes from a different spatial frequency due to aliasing. This effect can always be observed in a noise-free imaging system, but if the OTF drops off sharply enough, it may not be observed over noise sources in the system and therefore does not affect the recorded images. We hope this clarifies our comment in the manuscript.

“A step-by-step demonstration of preparing, building, aligning, and starting the extension setup enables widespread adoption.” I don’t understand what “starting the extension setup” means in this context, probably just needs to be reworded.

We have updated the sentence to read “A step-by-step demonstration of preparing, building, aligning, and initializing the hardware extension enables widespread adoption.”

“Our finalized setup enhances resolution, surpassing that of a traditional widefield microscope by factor of 1.55.” No, this is your theoretical extension the measurements demonstrated both more and less than this.

We have already discussed the apparent larger resolution increases, which are due to the difference between assessing resolution through two-point measurements versus spatial frequency support in a previous comment. We have clarified this sentence, which now reads “Our finalized setup enhances resolution, surpassing that of a traditional widefield microscope by a theoretical factor of 1.55.”

“This extension can be effortlessly affixed to microscope bodies with an epi-fluorescence port, enhancing its versatility. The accompanying software is accessible to users of all levels of expertise.” I think all levels of expertise is stretching the point a little. I think a more realistic assessment of the expertise required to both setup such a system and then for users to use it would definitely benefit the readers.

We thank the reviewer for the suggestion. We have changed the phrase slightly to state that a certain level of expertise is required to build the setup. Lines 488-491 now read “This extension can be easily attached to microscope bodies with an epi-fluorescence port, increasing its versatility. The accompanying software is intended for technically skilled biologists, lab technicians and tinkerers. Although we aim to lower the barriers to wider participation, especially for users from low-income countries, a certain level of technical knowledge is still required for setup and operation.”

Conclusion: Overall this is a good paper, and should be published with minor updates.

We thank the reviewer for the positive evaluation of our work.

Reviewer #2:

The authors propose an open-source design for a 2-color structured illumination module that attaches to commercial and open source fluorescence microscopes, providing a 1.55 fold improvement for a moderate cost of 3k. The openSIMMO is automated using open-source software and assembly is well documented to allow others to build their own module.

Although other SIM modules have been already been published, openSIMMO is unique because it integrates seamlessly with other existing open-source software and hardware, such as ImSwitch and openFrame. It also provides high illumination pattern contrast thanks to the use of coherent illumination, which results in increased resolution. For these reasons, I believe that this paper will be of great interest to others of the community and the wider field, and that many teams will choose to build one.

I believe the openSIMMO will be easy to reproduce thanks to the detailed online documentation provided by the authors, as well as their use of the git platform for users to ask for assistance if needed.

We thank the reviewer for the positive evaluation of our work.

[Major Comment]

In this paper, the authors are sharing the design and characterization of a new open-source structured illumination module. They provide a thorough characterization of the performances of the SIM module, and detailed instructions on how to build it. It is very appreciable that openSIMMO integrates with other open-source software and hardware, which helps with the maintainability of the whole open-source ecosystem, as well as facilitates user experience.

[Minor comments]

Title

1. consider “2 color” instead of “multicolor” for added clarity

We thank the reviewer for the suggestion and have changed the title to “Fully-Automated Multicolour Structured Illumination Module for Super-resolution Microscopy with two Excitation Colours” to account for the fact that we only use two excitation lines, but could use many more emission spectra depending on the dyes used here.

Introduction

2. [p1] “Structured illumination microscopy [...] only requires few photons” Could you provide citation/data to support this claim (or adjust the claim)

We have adjusted the claim to better reflect the advantage of SIM in live-cell imaging. Instead of stating that SIM requires fewer photons, we now emphasize that it operates with reduced excitation intensity, which is gentler on living cells. This adjustment aligns more accurately with the benefits of the technique in terms of minimizing photodamage during imaging.

“Structured illumination microscopy (SIM), as a widefield imaging technique, typically offers a maximum improvement in resolution by a factor of 2. It requires lower excitation intensity, making it a gentler method for imaging living cells, while keeping the amount of data necessary for the resolution improvement small. This makes it ideally suited for observing dynamic cellular processes with minimal photodamage.”

SIM achieves its high resolution by illuminating fluorescent samples with a series of very fine periodic patterns of very high contrast and detecting images at various pattern positions for computationally reconstruction.

We have made the suggested edit.

4. [p4] “Unlike the multi-color incoherent illumination of OpenSIM” suggested edit for clarity

We thank the reviewer for this suggestion and added a statement about the incoherent nature of OpenSIM: “Unlike the multi-color-based illumination of OpenSIM, which enhances existing commercial microscopes and relies on incoherent LED (light emitting diodes) illumination, our method not only offers improved image quality but also integrates open-source software.”

5. [p4] “The add-on includes both hardware and software components and is supported by extensive documentation (including parts lists and setup videos).” The documentation is indeed extensive. The authors clearly invested a lot of time and effort to make sure that their SIM module can be built by non-experts.

6. [p5] “We provide a SIM acquisition and reconstruction plugin for the Python-based open-source microscopy control software ImSwitch” It is great that the authors chose to integrate their microscope control with existing open source software, which will increase the maintainability and ease of use.

We thank the reviewer for their positive comments.

****Methods****

7. [p13] “Previous experiments have reported DMD mirror rotation parameters which differ somewhat from the nominal values [16, 26, 27]. To address this possibility, we infer the true mirror rotation parameters for the - state of our DMD by displaying a series of structured patterns on the DMD, recording the intensity of their various diffraction sub-orders on a camera, and fitting our model to the results” Given the manufacturing tolerance of the DMD, as well as the difference you observed, what do you expect in terms of impact on performance for openSIMMO?

We thank the referee for bringing up this important point. We have added this quantification to the paper, and we find that the effect of DMD manufacturing tolerances has a relatively small effect on the predicted quality of the SIM patterns, as quantified using their modulation contrast. Specifically, we now define the beam imbalance parameter η and detail its effect on the modulation contrast m in lines 168–170, which now read “We discuss this effect extensively in previous work, and find that an imbalance factor of $\eta \leq 1$ between the two beams causes the SIM modulation contrast to be degraded to $m = 2\eta/(1 + \eta^2)$ ”.

We quantify the pattern quality in lines 288–289, which now read “Using these values, we estimate $\eta = 0.64$ and 0.67 and maximum achievable modulation contrasts to be 0.91 and 0.93 for 488 nm and 635 nm respectively.”

8. [p13] “This reduces the alignment effort to a minimum, with assembly by semi-experienced opticians being possible in about 2 hours (provided that all parts are prepared in advance), based on detailed step-by-step documentation [37].” Ref 37 points to the UC2 website. I did not find the step by step documentation for this SIM module there.

We thank the reviewer to point out a wrong link in the reference. We have updated this to become <https://opensimmo.github.io>

9. [p15] first mention of openSIMMO as a name for the SIM module developed by the authors. Could you explain what the acronym stands for? You could also consider introducing it earlier in the paper for clarity.

We thank the reviewer for this valuable suggestion. We have introduced the synonyme "openSIMMO" (open-source SIM Module) in multiple places.

Results

10. [p18] "The finest line pair that can be resolved with the deconvolved widefield image is 210 nm." Could you indicate which method was used to do the deconvolution. This is important in the context of quantifying the resolution.

Indeed, this information was missing in the manuscript. The data was reconstructed with FairSIM. We added this information about the reconstruction method and parameter in the main text.

11. [p18] "This corresponds to a 1.75-fold improvement compared to the Abbe diffraction limit, whereby a 1.55-fold improvement would have to be expected. The difference could be due to the undersampling with 488nm laser excitation without using the in-built 1.5× intermediate magnification lens. For getting higher signal of the raw data, the magnification lens was removed to increase the photon number on each camera pixel." Could you clarify? When was the 1.5x magnification used/not used. For the wide field image? For the SIM image? As deconvolution and SIM reconstruction both depend a lot of the signal to noise, it is important for the reader to understand the exact experimental conditions. I also struggle to follow the reasoning regarding the 1.75x vs 1.55x resolution improvement.

We thank the reviewer for this important remark. For a detailed discussion of the 1.75× vs. 1.55× resolution enhancement, see our response to referee #1.

12. [Figure 4] Could you comment on the artifacts visible on the image and due to the SIM reconstruction and leftover illumination frequency. It might be interesting to experiment with the reconstruction parameters.

We thank the reviewer for this comment. We observed artifacts in the reconstruction, particularly in the low-magnification images (Figures 4b and 4d). These images were captured using a 20x/0.75 NA objective paired with a 6.5 μm pixel camera, which results in undersampling. This undersampling limits the reconstruction of high-frequency details, so fine structures in the sample aren't well-represented.

Discussion

13. Could you say a word about how the pattern spatial frequency would be adjusted depending on the nominal values of the objective used and how it is done in terms of software and hardware? This is important given that users might have objectives with different NA, magnification, etc.

This information was indeed missing in the manuscript. To promote reproducibility, we have added an explanation of how to generate the illumination pattern and create the Fourier mask.

14. Could you add a line about what would need to be changed to adapt to other fluorescence microscopes (ie they have different tube length focal distance and will need a difference mechanical attachment).

We have added the part numbers for our components respectively. Based on this we believe it will be possible to select alternative components based on the microscope stands existing in people's labs.

We have updated lines 301–305 to: Using different adapter plates (e.g. Thorlabs SM1A30 for the Nikon Ti2) and tube lenses (e.g. Thorlabs AC254-200-A-ML, $f' = 200\text{ mm}$ for the Nikon Ti2) respectively enables the adaptation to different microscopy bodies.

****Online resources**** <https://opensimmo.github.io/docs/>

15. “The module provides an enhancement factor of 1.75x compared to general widefield microscopy.” Is it 1.55 or 1.75? this contradicts the value given in the paper

Thanks for the comment. We have updated this statement to clarify that 1.55 is the correct enhancement factor and included the DOI of this manuscript’s preprint from Biorxiv.

16. It is great that the authors included detailed instructions regarding laser safety. The explanations are detailed, illustrated with an abundance of pictures and several videos.

We thank the referee for this positive comment.

17. “Aligning the Fourier-filtering telescope → The CAD files will come soon (file an issue or write use if we forgot - sorry!)” Could the files be added?

Thanks for pointing this out, we have updated the website with the files accordingly.

Reviewer #3:

In this work, Wang et al. presented an open-source, fully-automated, two-color structured illumination module (openSIMMO) which can achieve up to a 1.55-fold improvement in lateral resolution. A model was developed to ensure optimal diffraction performance using a DMD with tilted and rolling pixels, thereby extending the potential use of low-cost video projectors for SIM setups. This is complemented by providing comprehensive open-source documentation and a modular software framework compatible with various hardware components (e.g., cameras, stages) and reconstruction algorithms. Overall, this work provides a highly impactful approach for open-source super-resolution microscopy, by making SIM more accessible to a broader range of microscopists and biologists.

There are some issues that the author needs to consider and address before the manuscript can be published. Major Comments: 1. The author claims that the system can achieve real-time super-resolution imaging, yet the article does not explicitly state the imaging speed of the system. To my knowledge, the speed of a microscopic system is related to both hardware and software. On the hardware side, it is related to the performance of devices such as the DMD and camera. It is suggested that the author provides a sequence diagram. On the software side, the integration of hardware and algorithms are achieved by using ImSwitch. However, when ImSwitch facilitates communication between different hardware and synchronizes various devices, it may introduce additional latency that could reduce imaging speed. It is hoped that the author can provide a more detailed discussion on this matter.

We thank the reviewer for this valuable comment. We have added an additional sequence diagram in the supplementary figure 5 that describes the possible limitations and the overall framework of the acquisition workflow:

“The trigger diagram illustrates the temporal coordination between various devices and image processing routines. The primary bottleneck in imaging speed is the camera sensor’s exposure time, $t_{\text{exposure}} > 50\text{ ms}$,

largely due to the relatively low laser power reaching the sample. To avoid additional delays in the acquisition process, frame acquisition (i.e., transferring data from the camera to RAM) and image processing are handled asynchronously. The Digital Micromirror Device (DMD), controlled by a Raspberry Pi, serves as the master trigger, dictating the camera's frame acquisition. ImSwitch monitors the availability of the framebuffer and manages system parameters, including toggling the laser, pausing experiments, and handling image stack reconstruction and display."

2. The author should provide a more detailed discussion on the impact of the blaze condition violation on the illumination pattern of SIM. Since the output angle of diffraction orders are wavelength-dependent, while the blaze envelope is wavelength-independent, when the DMD is used for two-color imaging, aligning the optical path may result in one beam deviating more from the blaze condition. It is recommended that the author provide a quantitative analysis of the blaze condition violation and illumination pattern's contrast (or brightness), to offer criteria explaining within what range the blaze condition violation is acceptable.

We thank the referee for this important observation.

We have already modelled and discussed the impact of blaze condition violations extensively in our previous work "Multicolor structured illumination microscopy and quantitative control of polychromatic light with a digital micromirror device," *Biomed. Opt. Express* 12, 3700–3716 (2021). We clarify this point in the text by editing lines 167–169 to read "We discuss this effect extensively in previous work, and find that an imbalance factor of $\eta \leq 1$ between the two beams causes the SIM modulation contrast to be degraded to $m = 2\eta/(1 + \eta^2)$."

Additionally, we now provide a quantification of how the blaze condition violation will affect the beam imbalance and modulation contrast for several different possible alignment choices. Lines 266–269 now read "In this case, using the $1.55\times$ patterns discussed in section S4 will lead to $\eta = 0.95$ and 0.86 for the 488 nm and 635 nm wavelengths respectively. In either case, the effect on the modulation contrast is small, resulting in $m \geq 0.99$." Lines 277–279 now read "In this case, using the $1.55\times$ patterns leads to $\eta = 0.94$ and 0.71 , leading to modulation contrasts of 0.999 and 0.94 for 488 nm and 635 nm respectively." Lines 288–289 now read "Using these values, we estimate $\eta = 0.64$ and 0.67 and maximum achievable modulation contrasts to be 0.91 and 0.93 for 488 nm and 635 nm respectively."

We note that the acceptable blaze condition violation depends on the choice of SIM resolution enhancement, as patterns with more aggressive resolution enhancement will result in smaller values of η and lower modulation contrasts.

3. Please indicate the position of the AC-254-200 lens in the system diagram of Figure 1 and FIG S6.

We thank the review for this valuable comment. We have included the indicator for the tube lens in both figures.

4. FIG 3a presents the graphical user interface of ImSwitch, which includes available widgets such as View, Recording, Laser, and so on. However, these widgets have already existed in the original ImSwitch [1] and are common parts adapted to any microscopic system. As the control software for openSIMMO used to achieve SIM image acquisition and reconstruction, it is recommended that the author include content related to custom plugins specific to openSIMMO, such as showcasing the SIM widget in the software interface screenshots, and demonstrating the processing of SIM raw images through ImSwitch and the display of reconstruction results using Napari. An open-source software is highly welcome to facilitate the community to follow this work.

We have updated the screenshot to reflect the ability to use SIM in ImSwitch accordingly. Thank you.

5. FIG 3b exhibits noticeable periodic stripe-like artifacts, while FIG 3c shows evident honeycomb artifacts in mitochondrial reconstructions, likely due to defocused background. FIG 4b also displays prominent honeycomb and hammerhead artifacts, which interfere with the accurate assessment of the sample's fine structures. These artifacts may have been introduced for several reasons:

[1] Low signal-to-noise ratio (SNR) during image acquisition. It is recommended to increase the exposure time or enhance the illumination power on the sample.

[2] The Fourier transforms of the SIM images FIG S2 and S3 contain conspicuously bright spots in the spectrum. It is advised to use algorithms such as OPEN-SIM2 or HIFI-SIM3 for reconstruction, as these algorithms are more robust and less sensitive to system parameters like PSF, significantly reducing spectral anomalies and eliminating artifacts caused by defocused background.

Additionally, these bright spots might be caused by incomplete filtering of stray light. It is recommended that the authors verify the optical path to prevent unnecessary diffraction orders or stray light from illuminating the sample.

We thank the reviewer for his suggestions. We tried to use the HIFI-SIM algorithm from the publication [nature.com/articles/s41377-021-00513-w](https://doi.org/10.1038/s41377-021-00513-w), but it showed some errors in the reconstruction. We will test the algorithm again and put some information on the tutorial website. We have chosen Napari-SIM inside our data acquisition software ImSwitch, since it was straight forward to implement and the ability to use it for fast (e.g. online) reconstructions. The bright spots could be caused by unwanted diffraction orders passing through the Fourier filter. This can be prevented by using a different illumination pattern and a more precise Fourier filter mask.

6. The authors claimed that “to promote low-cost access to SIM imaging beyond previous methods, I have demonstrated a fully open-source dual-color (488/635 nm) structured illumination microscopy (SIM) add-on module, which can be easily adapted to commercially available fluorescence microscopes.” However, numerous open-source algorithms and hardware solutions for SIM have already been published, including related work previously developed and mentioned by the author in the introduction. The author has provided the specific components and costs used in this work in the supplementary materials (Table S3). It is recommended that the author provide a table comparing the costs and achievable functionalities of previous open-source hardware with those of this work, in order to highlight the significant improvements in cost-effectiveness or performance over previous works.

We have included the publications in our reference. We have looked at various open-source SIM configurations, but most of them have not provided a bill of materials of the components implemented in their hardware. It is difficult to make a direct comparison of the hardware costs of different approaches.

7. The authors claimed that “This contrast, although reduced, still surpasses that achievable with incoherent SIM, thereby enhancing resolution reconstruction.” However, since theoretically only a 1.55-fold resolution improvement can be achieved, the signal-to-noise ratio or contrast with incoherent illumination would not degrade as severely as it would with a 2-fold resolution. Although the author has used beads to demonstrate the pattern contrast, the significant variation across different angles is not convincing enough. It is recommended that the author provide simulations and theoretical calculations to quantitatively compare the effects of incoherent illumination and the polarization-induced interference patterns used in this study.

We agree with the reviewers point that for the moderate resolution enhancement factor used in this work, incoherent SIM will still provide reasonable modulation contrast and resolution enhancement. As such, we have removed the sentence comparing our implementation of coherent SIM without polarization control with incoherent SIM to avoid confusion.

8. The authors choose the higher order of diffraction of DMD to conduct 2DSIM, while the traditional 2DSIM uses the ± 1 st. I wonder about the photon intensity loss of this schedule. How is the intensity ratio between the ± 1 st order?

We disagree with the reviewer's claim that most 2D SIM instruments using DMD's rely on the ± 1 st diffraction orders. There are two types of diffraction orders that must be distinguished here, the diffraction orders off of the underlying mirror array, and diffraction orders from the pattern which is displayed on those mirrors. The higher diffraction orders we use, $(-6, 0)$ and $(-5, 0)$, refer to diffraction off the underlying mirror array. Similar orders are used in other DMD work (e.g. the work by Sandmeyer et al. cited later by the referee). When a pattern is displayed on the DMD, this generated sub-diffraction orders about the main diffraction order. Often the sub-diffraction orders associated with the SIM pattern are referred to as the ± 1 order. In this work, we use the ± 1 suborders as is typical.

9. In Fig. S2 and S3, the cell is too thick, not suitable for 2DSIM imaging. Due to the visible artifacts, I suggest authors choose a thin sample for display.

We think that the artifacts are not due to the thickness of the sample. The samples shown in Fig. S2 and S3 are stained BPAE cells, which have a small thickness. The reduced image quality is limited by the low quantum efficiency of the industrial camera and unwanted diffraction orders that may still pass the Fourier filter as they are relatively close to the zeroth order. It can be improved by using a scientific camera with better illumination patterns.

10. Comparing with 2DSIM, 3DSIM can further improve the axial resolution. It has been widely applied in cell research. I hope that the authors can discuss the possibility of improving OpenSIMMO into 3DSIM in the future in the discussion part.

We have added a discussion of experimental changes necessary to adapt openSIMMO for 3D SIM with 3-beams. Please refer to our response to referee #1.

11. The following open-hardware developments are for the authors' reference:

- Liu, Q., Zhou, D., Zhang, J., Ji, C., Du, K., Chen, Y., ... & Kuang, C. (2023). DMD-based compact SIM system with hexagonal-lattice-structured illumination. *Applied Optics*, 62(20), 5409-5415.
- Sandmeyer, A., Lachetta, M., Sandmeyer, H., Hübner, W., Huser, T., & Müller, M. (2019). DMD-based super-resolution structured illumination microscopy visualizes live cell dynamics at high speed and low cost. *BioRxiv*, 797670.
- Li, Y., Cao, R., Ren, W., Fu, Y., Hou, Y., Zhong, S., ... & Xi, P. (2024). High-speed autopolarization synchronization modulation three-dimensional structured illumination microscopy. *Advanced Photonics Nexus*, 3(1), 016001-016001.
- Xu, X., Wang, W., Qiao, L., Fu, Y., Ge, X., Zhao, K., ... & Xi, P. (2024). Ultra-high spatio-temporal resolution imaging with parallel acquisition-readout structured illumination microscopy (PAR-SIM). *Light: Science & Applications*, 13(1), 125.

We would like to point out that the Sandmeyer paper is our reference 26. We have added citations to the other works where appropriate.

Minor comments: 1. The author should consider adding line numbers.

We have added line numbers to the new version of our manuscript

2. The acronym "openSIMMO" lacks its full name when it first appears in the text.

We now introduce this acronym where it is first discussed.

3. Line 2 Page 11: “R23=-R23” appears to be a typographical error for “R13=-R23”.

We have corrected this typo.

4. Line 5 Page 13: The term “nmorder” is confusing; perhaps there is a missing space that should be between “nm” and “order”.

We have corrected this typo.

Reference

1. Casas Moreno, Xavier, et al. “ImSwitch: Generalizing microscope control in Python.” *Journal of Open Source Software* 6.64 (2021).
2. A. Lal, C. Shan, P. Xi, “Structured Illumination Microscopy Image Reconstruction Algorithm,” *IEEE Journal of Selected Topics in Quantum Electronics*, vol. 22, no. 4, pp. 50-63, July-Aug. 2016, Art no. 6803414.
3. Wen, G., Li, S., Wang, L. et al. “High-fidelity structured illumination microscopy by point-spread-function engineering.” *Light Sci Appl* 10, 70 (2021).

Dear Editor,

We are grateful for the positive feedback from the reviewers and are pleased that the manuscript is now considered suitable for publication.

In response to the minor comment from **Reviewer #1**, we have clarified the wording to ensure the meaning is fully understood. The revised sentence reads:

"A step-by-step demonstration guides users through preparing the components, building and aligning the hardware setup, and configuring the system for operation, ensuring accessibility and ease of adoption."

This change explicitly communicates that we provide guidance for preparing, assembling, aligning, and running the imaging system.

We appreciate the reviewers' constructive feedback, which has helped improve the clarity of our work, and thank the editorial team for facilitating the review process.

Sincerely,
Benedict Diederich